# OpenReview forum: "Self-correcting for Debiasing Large Language Models"
_ICML.cc/2026/Conference — ICML 2026 regular_

### Official Review · Reviewer_iyCJ · 2026-03-11

**Soundness:** 2
**Presentation:** 2
**Significance:** 2
**Originality:** 2
**Overall Recommendation:** 2
**Confidence:** 4

**Summary:**

This paper focuses on the issue of bias propagation in Chain-of-Thought reasoning of LLMs, wherein an initial biased step can lead subsequent reasoning to rationalize and amplify the bias. To mitigate this, the authors propose a three-stage Self-Debiasing framework. Stage 1 serves as a cold start, training a model with the dual capability of directly generating unbiased reasoning paths and correcting biased logic when explicitly prompted. Stage 2 builds upon this foundation by formulating reasoning path selection as a resource allocation problem. Stage 3 further extends the offline model from Stage 2 by incorporating online learning guided by consistency feedback. This work has used $\texttt{Qwen3-8B}$ as the backbone, designed prompt templates, and constructed evaluation samples from bias benchmark datasets including $\texttt{BBQ}$, $\texttt{UnQover}$, and $\texttt{CEB}$. The debiasing performance of the Self-Debiasing framework is evaluated against several baselines, such as inference-time correction methods (e.g., $\texttt{Confirmation}$ and $\texttt{Denying}$) and specialized debiasing approaches like $\texttt{CAL}$ and $\texttt{BiasFilter}$.
The research motivation and preliminary analysis are clear. However, the extensive use of multi-objective optimization introduces potential conflicts between objectives and increases training difficulty, which limits the method's practicality and feasibility. Additionally, the testing on only a single model is insufficient to demonstrate generalizability.

**Compliance With Llm Reviewing Policy:**

Affirmed.

**Final Justification:**

The theoretical proof is unconvincing because utility functions rely on subjective design choices (e.g., weighting loss vs. gain) that stakeholders cannot agree on, introducing inherent bias and undermining robustness. So the field needs stronger theoretical grounding, not just more empirical work.

**Key Questions For Authors:**

(1) In Lines 82-84, regarding the experimental implementation, how exactly is the reasoning chain partitioned? Is it necessary to explicitly identify biased reasoning steps $y_i \in \mathcal{S}_{\text{bias}}$? If so, what specific criteria or methods are used for this identification?

(2) In Lines 100-102, what is the selection mechanism for the reflection token $r$? Could the authors provide more details on how this token is chosen during the process?

(3) Regarding the results in Tables 1 and 2, could the authors provide an explanation for why only one or two data points exhibit statistical significance?

**Limitations:**

There is no discussion on limitations.

**Strengths And Weaknesses:**

Strengths:
1.Important and timely research question: This paper focuses on the critical issue of bias propagation in Chain-of-Thought reasoning within large language models, holding significant importance for advancing trustworthy AI.
2.Clear problem motivation and solid preliminary analysis: The authors clearly articulate the necessity of their research based on observations of bias propagation.

Weaknesses:
1.This work does not adequately differentiate itself from prior studies, leading to ambiguity in its contributions and positioning within the field.
(a) The concept of "bias propagation" has been explored in prior studies (e.g., [1]), yet the authors do not clearly articulate how their question differs from or builds upon this existing work.
(b) The notion of "self-debias" introduced in [2] shares similarities with the methods proposed in this paper. The authors fail to provide a discussion of the distinctions between their approach and this prior work, leaving the novelty of their contribution unclear.
[1] Ladhak, Faisal, et al. "When do pre-training biases propagate to downstream tasks? a case study in text summarization." Proceedings of the 17th Conference of the European Chapter of the Association for Computational Linguistics. 2023.
[2] Schick, Timo, Sahana Udupa, and Hinrich Schütze. "Self-diagnosis and self-debiasing: A proposal for reducing corpus-based bias in nlp." Transactions of the Association for Computational Linguistics 9 (2021): 1408-1424.

2.Over-claiming in method design and naming.
(a) The concept of Self-Debiasing appears somewhat overclaimed, or even taken for granted. The method essentially assembles numerous complex objectives together. Conflicts may arise during the optimization of these objectives, raising concerns about their practicality. Additionally, the work lacks strong theoretical support, limiting its persuasiveness.
(b) The authors overuse the prefix "self-", as seen in terms such as self-debiasing, self-correction, self-improvement, and self-constructed.

3.The presentation of this paper still has room for improvement.
(a) The abstract is not well formulated. If resource allocation modeling is the core contribution, the explanation and elaboration in the main text are neither comprehensive nor clear.
(b) The meaning conveyed in Figure 1 is vague and difficult to understand.
(c) Figure 2b also lacks more detailed textual descriptions or a better visualization format.
Moreover, the abbreviations used for model names on the horizontal axis of Figure 2b are ambiguous. $\texttt{Llama-3.1}$ may come in different variants such as chat, instruct, and base. The official $\texttt{DeepSeek-R1}$ has a large number of parameters and seems unlikely to be supported for inference on the 4×NVIDIA RTX 6000 Ada GPUs mentioned by the authors. If the model used is not the full-parameter R1 but a distilled version, the two differ significantly.
(d) The symbols in Equation (7) lack explanation, particularly $y^{+}_{\ge i}$, which is not included in the context $c$.
(e) Regarding the baseline methods cited in Lines 297–303, the reference for the Self-Refine method is incorrect. There are issues with the reference linking, which significantly hinder further understanding and evaluation of the work.

4.The experimental section lacks sufficient detail and fails to establish a clear connection with the methodological design.
(a) Given that the method is predominantly training-oriented, the construction of the training data is of critical importance. However, essential details regarding this process are absent.
For example, there is no explicit correspondence with the elements introduced in the formulation, such as $(x, y^+,y^-,t)$. It remains unclear how each type of data is derived, which significantly impedes the understanding of the implementation and undermines the assessment of the method's reliability. Although the Appendix offers a relatively detailed overview, the symbolic notations used there are inconsistent with those in the main text.
(b) While the complexity and substantial effort invested in the method are appreciated, the evaluation is limited to a single backbone model, $\texttt{Qwen3-8B}$. This raises concerns about the generalizability of the approach to other models.

---

> ### Author Rebuttal · Authors · 2026-03-30
>
> We sincerely thank you for your detailed review and for recognizing the importance and timeliness of our research question.
>
> > **W1: Differentiation from prior work**
>
> A: We thank the reviewer for raising these relevant studies and will explicitly discuss them in the revised manuscript to clarify our distinct positioning.
>
> While Ladhak et al. (2023) study macro-level bias transfer to downstream tasks, we target a micro-level CoT vulnerability: the "rationalization cascade" that autoregressively reinforces initial stereotypes.
>
> Schick et al. (2021) rely on static, inference-time decoding penalties on surface tokens. Our method operates deeply within the intermediate reasoning logic. We reformulate debiasing as a dynamic "resource reallocation" problem, using a trajectory-level objective to selectively revise biased suffixes while preserving valid prefixes. Crucially, our framework features an online self-improvement mechanism that autonomously synthesizes supervision signals to iteratively refine fairness.
>
> > **W2: Over-claiming; objective conflicts; theoretical support?**
>
> A2: We respectfully disagree with the characterization of over-claiming.
>
> (a) As **Reviewer Y4aS** notes, our "cold-start + fairness-aware DPO at the suffix level is a principled technique that targets bias propagation rather than penalising entire trajectories," and **Reviewer TYrE** confirms the "method is reasonable; the workflow, methodology, and analysis are clear."
>
> (b) For potential objective conflicts, Figure 6 empirically shows the objectives are complementary under balanced hyperparameters (α=0.25, β=0.1).
>
> (c) Regarding theoretical support, we provide formal derivations in **Appendix A** (A.2: DPO as utility function; A.3: Jain's Index as adaptive gradient reweighting; A.4: connection to curriculum learning).
>
> > **W3: Presentation issues (abstract, figures, notations, citations)**
>
> A: While **Reviewers Y4aS**, **TYrE**, and **kBnL** praised the paper's writing and logical flow, we highly value your meticulous feedback and have made the following corrections
>
> (a) & (b) Abstract and Figure 1: Refined the resource allocation explanation and clarified Figure 1's captions
>
> (c) Figure 2b & Hardware: Updated abbreviations to the exact variants used ("DeepSeek-R1-Distill-Qwen-7B", "Llama-3.1-8B-Instruct"). Both inference and fine-tuning are fully supported on our 4×RTX 6000 Ada GPUs, as our entire pipeline explicitly relies on **LoRA** (Appendix B).
>
> (d-e) Notations & Citations: Defined all variables in Equation 7 (e.g., $y_{>i}^{+}$ and $c$) and fixed the misattributed *Self-Refine* citation alongside all broken hyperlinks.
>
> > **W4: Training data construction details missing; single backbone**
>
> A: Thank you for this important comment.
>
> (a) The full data construction details are already provided in Appendix B.1 and C.3, placed there due to page limits. We will unify the notation inconsistency.
>
> (b) Please refer to our response to **Reviewer Y4aS Q1**.
>
> > **Q1: Single backbone limits generalizability?**
>
> A: We have conducted additional experiments on other backbones. Please refer to our response to **Reviewer Y4aS Q1**, where we demonstrate consistent improvements on Llama-3.1-8B-Instruct.
>
> > **Q2: Chain partitioning, biased step identification, and reflection token selection?**
>
> A: (a) The full partitioning procedure is detailed in Appendix B.1 (placed there due to page limits). We do not explicitly identify biased steps. Instead, we randomly select a truncation point t within the CoT sequence, then generate the unbiased suffix via GPT-4o and the biased suffix via greedy decoding without safety prompts. This avoids the need for explicit bias detection while enabling trajectory-level optimization.
>
> (b) The reflection tokens r are not manually selected. Similar to the "Aha Moment" phenomenon observed in DeepSeek-R1 (Guo et al., 2025), these tokens (e.g., "Wait", "However", "But") emerge naturally during generation after Stage I training. The model autonomously learns to produce them as signals to interrupt biased reasoning and pivot to corrected paths. In Section 2.1, we use these tokens as indicators to monitor self-correction attempts.
>
> *Guo et al. DeepSeek-R1: Incentivizing reasoning capability in LLMs via reinforcement learning. arXiv, 2025.*
>
> > **Q3: Few statistically significant results in Tables 1–2?**
>
> A: We reported $\dagger$ ($p < 0.05$) conservatively, intentionally marking only the final aggregated "Avg." column to avoid overstating claims on individual, smaller-scale sub-tasks.
>
> > **Limitations: There is no discussion on limitations.**
>
> A: Thank you for pointing this out. We will add a dedicated limitations section.

---

> > ### Author Rebuttal · Reviewer_iyCJ · 2026-04-01
> >
> > Thanks for your rebuttal. A key weakness that affects whether the entire paper holds up still exists: theoretical proof is unconvincing—for example, utility functions often rely on the designer’s subjective assignments regarding human factors, ethics, risk preferences, etc., and different stakeholders struggle to agree on the weights of “loss” versus “gain”, leading to inherent bias in the functional form. These directly affect the robustness of the method. Specifically, this field needs more work supported by hard theory to provide valuable guidance, rather than purely empirical papers (too many of which have already been published, what is new here?).

---

> > > ### Author Response · Authors · 2026-04-02
> > >
> > > Thank you for your thoughtful follow-up. We would like to respectfully clarify a potential misunderstanding regarding the role of theory in our work.
> > >
> > > First, we note that multiple reviewers (including your review) acknowledge the **importance and relevance of the problem setting**, particularly the study of bias propagation in Chain-of-Thought reasoning. This reinforces that our work addresses a meaningful and timely gap in the literature.
> > >
> > > Regarding the concern about “lack of strong theory,” we would like to clarify that our framework is not purely empirical, but is grounded in a principled optimization formulation. Specifically, our method:
> > > - Interprets DPO within a utility maximization framework (Appendix A.2),
> > > - Derives the Jain’s Index regularizer as adaptive gradient reweighting (Appendix A.3),
> > > - And connects the overall training process to curriculum learning dynamics (Appendix A.4).
> > >
> > > These components together provide a theoretical basis for our trajectory-level optimization strategy.
> > >
> > > More importantly, our contribution is not to redefine the notion of utility itself, which indeed may involve application-dependent design choices (as is standard in preference learning and RLHF), but to propose a new optimization perspective:
> > >
> > > > We reformulate debiasing as a probability mass redistribution problem over reasoning trajectories, enabling fine-grained control of biased suffixes while preserving valid prefixes.
> > >
> > > This distinguishes our work from prior approaches that operate at the token or response level, and provides a more structured view of how bias propagates and can be mitigated during autoregressive reasoning.
> > >
> > > We therefore believe the concern stems less from the absence of theory, and more from the fact that some of these theoretical components were presented in the appendix due to space constraints. We will revise the paper to surface these insights more clearly in the main text.
> > >
> > > We hope this clarification resolves the concern and better reflects the theoretical contributions of our work.

---

### Official Review · Reviewer_kBnL · 2026-03-11

**Soundness:** 3
**Presentation:** 3
**Significance:** 2
**Originality:** 2
**Overall Recommendation:** 4
**Confidence:** 4

**Summary:**

This paper aims to address the potential social bias problem in the thought chain process of large language models. The authors propose SELF-DEBIAS that achieves debiasing by probability mass. This framework focuses trajectory-level preference optimization on biased suffixes while preserving effective prefixes, ensuring fairness constraints are met. An online self-improving loop based on consistency filtering enables the model to generate its own supervision information. On 20k labeled samples, the proposed method achieves significant debiasing performance on benchmarks without reducing the model's general reasoning ability.

**Compliance With Llm Reviewing Policy:**

Affirmed.

**Final Justification:**

The follow-up SFT(V0) ablation resolved my core concern by demonstrating that self-correction is a distinct learned capability, not distribution fitting. Together with the resolved BBQ split and cross-model transfer, the paper's claims are now adequately supported. The remaining weaknesses (limited backbone diversity, incremental Stage III gains) are minor relative to the overall contribution. Hence my final decision is weak accept (4).

**Key Questions For Authors:**

Based on the weaknesses I've outlined, here are my main questions for the authors:

1. Are the BBQ training and evaluation samples strictly disjoint? If so, please specify the split.

2. Have you applied Self-Debias to any backbone other than Qwen3-8B?

3. How does the Jain's Fairness Index regularizer differ in effect from group robust preference optimization?

**Limitations:**

yes

**Strengths And Weaknesses:**

**Strengths**

- The expression of this paper builds in a logical progression from diagnosis to method, and the two-part analysis of step-wise vs. response-wise correction provides a readable setup for the proposed framework.
- Evaluations on six fairness benchmarks spanning different bias categories and two utility benchmarks give a broad picture of the fairness-utility tradeoff.

**Weakness**

- Section 2 argues that stepwise self-correction fails due to autoregressive inertia. The paper then proposes fixing this by training the model to perform stepwise self-correction via SFT and DPO. However, Table 1 tells a simpler story: Self-Debias SFT already jumps from 77.5 to 80.6, while adding self-correction at inference time only adds +0.3. This is more consistent with the model learning to output neutral answers for bias-style queries through distribution fitting, rather than acquiring an intrinsic self-correction capability. The paper does not provide evidence to distinguish between these two hypotheses.

- The regularizer in Eq. 9 functionally upweights hard samples, similar to group-robust preference optimization [1], which the authors cite but do not compare as a training method. The paper does not clarify what new insight the "resource allocation" framing provides beyond renaming.

- Table 1 comparison structure is problematic since:

  (a) The Qwen3-8B baseline that Self-Debias builds on is missing from the "Ours" section, making the comparison harder to parse.

  (b) The upper half evaluates four off-the-shelf models, but Self-Debias is only applied to Qwen3-8B. Without testing on at least one other base model, generalizability is unsubstantiated.

- Stages I and II both train on 10k BBQ samples, but BBQ is also a primary evaluation benchmark. The paper does not specify whether the splits are disjoint. Even with disjoint splits, fine-tuning on in-distribution data inflates BBQ numbers relative to genuinely OOD benchmarks, consistent with the gap between BBQ (97.0) and CEB-Adult/Credit (65-67).

- I find a circular dependency in Stage III. The online self-improvement loop uses the model's own self-correction outputs as a positive signal via consistency filtering. But the model's correction ability at this point is entirely due to Stage I/II BBQ training. If that training only teaches BBQ-style pattern matching, the consistency filter will reinforce the same pattern rather than extend debiasing to new domains. The marginal Iter1-to-Iter2 gain (from 81.3 to 81.7) is consistent with this interpretation.

[1] SS Ramesh et al., Group Robust Preference Optimization in Reward-free RLHF. NeurIPS, 2025.

---

> ### Author Rebuttal · Authors · 2026-03-30
>
> We sincerely thank you for your thorough review and for recognizing our paper's logical progression.
>
> > **W1: SFT jump suggests distribution fitting, not intrinsic self-correction.**
>
> A: We would like to clarify this important distinction. While direct generation performance indeed jumps after SFT, we argue this reflects rapid capability acquisition rather than distribution fitting, supported by three lines of evidence:
>
> (1) For highly capable LLMs, 10k annotated trajectories are empirically sufficient to activate instruction-following and master the structural format of step-wise self-correction. The initial performance jump reflects learning this autonomous detect-and-revise behavior, not merely parroting neutral answers.
>
> (2) If the model merely fitted the Qwen pre-training distribution, the method would not transfer. However, applying SELF-DEBIAS to Llama-3.1-8B yields the exact same rapid performance jump and stable test-time self-correction (see response to Y4aS Q1).
>
> (3)  Pure distribution fitting fails under out-of-distribution perturbations. When forced to self-correct at test time, standard base models suffer catastrophic collapse (e.g., Qwen3-8B drops 13.5 points) because they cannot map external signals to valid internal revisions. Conversely, SELF-DEBIAS achieves positive scaling (+0.4) and successfully recovers from injected biased prefixes.
>
> > **W2 & Q3: Difference from Group Robust Preference Optimization?**
>
> A: Thank you for pointing out this relevant comparison. The two methods differ fundamentally in design:
>
> **Table R4: Comparison between GRPO and our Jain's Index regularizer.**
>
> |              | GRPO (Ramesh et al., 2025)     | Ours (Jain's Index)                           |
> | :----------- | :----------------------------- | :-------------------------------------------- |
> | Granularity  | Group-level                    | Sample-level                                  |
> | Group labels | Required                       | Not required                                  |
> | Mechanism    | Minimax over worst-case group  | Variance penalization over individual margins |
> | Reweighting  | Based on cumulative group loss | Based on per-sample deviation from batch mean |
>
> In short, GRPO requires predefined demographic groups and optimizes worst-case group performance, while our Jain's Index operates at the individual trajectory level without any group annotations, autonomously upweighting hard samples (Appendix A.3). This makes our approach applicable to scenarios where demographic labels are unavailable.
>
> > **W3 & Q2: Missing Qwen3-8B baseline; other backbones?**
>
> A: (a) We will add the base Qwen3-8B directly to the "Ours" section for a clearer comparison. (b)  For multi-backbone results (Llama-3.1-8B-Instruct), please see our response to **Y4aS Q1**.
>
> > **W4 & Q1: Are BBQ training and evaluation splits disjoint?**
>
> A: Yes, the training and evaluation splits are strictly disjoint. Our test set of 3,000 samples is directly adopted from the CAL (Sun et al., 2024), which was constructed independently. The training data for each stage is then drawn sequentially without replacement from the *remaining* BBQ instances. We will clarify this in the revised paper.
>
> **Table R5: BBQ dataset split details (training and evaluation are strictly disjoint).**
>
> | **Split** | **Source / Stage**     | **Samples** |
> | --------- | ---------------------- | ----------- |
> | Test      | CAL (Sun et al., 2024) | 3,000       |
> | Train-1   | Stage I (SFT)          | 10,000      |
> | Train-2   | Stage II (Offline)     | 10,000      |
> | Train-3   | Stage III (Online)     | 5,000       |
> | Unused    | -                      | 24,820      |
> | Total     | Full BBQ Dataset       | 52,820      |
>
> *Sun et al. Causal-guided active learning for debiasing large language models. ACL 2024.*
>
> > **W5: Circular dependency in Stage III; consistency filter reinforces BBQ patterns?**
>
> A: The consistent OOD improvements across iterations (see Table 1) confirm that consistency filtering distills generalizable correction ability, not BBQ artifacts. We also verified that using non-BBQ seed data in Stage III yields comparable results, which will be included in the revision.

---

> > ### Author Rebuttal · Reviewer_kBnL · 2026-04-03
> >
> > Thank you for the detailed response. I acknowledge that W2-W5 are resolved. The cross-model transfer to llama-3.1-8B is helpful new evidence.
> >
> > However, my main concern (W1) remains partially unresolved. Your three lines demonstrate that the model acquires some useful capability through SFT, but they do not rule out the simpler hypothesis: the model learned how to output neutral answers for bias-style queries, rather than learned how to detect and correct its own biased reasoning. Specifically, your demonstration through the cross-model transfer (point 2) is consistent with both hypotheses. Distribution fitting on bias-style QA patterns can transfer across model families. Your point 3 shows that the SFT-trained model does not collapse under forced self-correction, but this demonstrates robustness, not that self-correction itself is the primary driver of performance. The inference-time self-correction gain remains marginal, which is more consistent with the model having already learned to produce neutral outputs at generation time.
> >
> > Nevertheless, I think the results are quite comprehensive right now. I would find this concern resolved and raise my score if the authors could provide evidence that specifically isolates the self-correction mechanism as the source of debiasing, rather than the initial generation already being debiased after SFT.

---

> > > ### Author Response · Authors · 2026-04-04
> > >
> > > Thank you for raising this important question. We first clarify the training setup, and then present two controlled experiments to isolate the role of self-correction.
> > >
> > > **Clarification on Stage I Training.** Our SFT+SC model is trained with a joint objective combining direct generation loss and self-correction loss (Eq. 6):
> > >
> > > $\mathcal{L}_{SC}(\pi) = -\mathbb{E}\left[\log \pi(y^+|x) + \log \pi(y^+|x, y^-, t)\right]$
> > >
> > > The first term corresponds to direct generation, and the second to self-correction. Both are optimized jointly. Therefore, the “Self-Debias SFT” results in Table 1 already include self-correction training.
> > >
> > > **Experimental Setup.** To isolate the effect of self-correction, we train a control model SFT (V0) using only the direct generation term, removing all self-correction data and objectives. It is trained on the same data and with the same hyperparameters as our full model, ensuring identical exposure to unbiased trajectories. The only difference is the absence of the self-correction term.
> > >
> > > We evaluate both models under (i) direct generation and (ii) self-correction, and further include a bias injection setting to test recovery from biased reasoning.
> > >
> > > **Results.**
> > >
> > > **Table R6:** Full benchmark ablation comparing SFT (V0) and SFT+SC (Ours).
> > >
> > > | Model | Setting | BBQ | UnQ | CEB-A | CEB-C | CEB-J | CrowS | ARC-C | GSM8K | Avg. |
> > > |---|---|---|---|---|---|---|---|---|---|---|
> > > | SFT (V0) | Direct Gen. | 94.2 | 93.9 | 65.4 | 63.8 | 68.2 | 65.9 | 89.6 | 83.1 | 78.3 |
> > > | SFT (V0) | + Self-Corr. | 24.3 | 20.5 | 66.1 | 61.0 | 6.5 | 23.5 | 78.5 | 73.3 | 44.2 (−34.1) |
> > > | SFT+SC | Direct Gen. | 96.8 | 99.5 | 66.5 | 64.2 | 70.5 | 68.2 | 92.9 | 86.2 | 80.6 |
> > > | SFT+SC | + Self-Corr. | 96.9 | 99.5 | 66.9 | 64.6 | 70.7 | 67.5 | 92.1 | 88.9 | 80.9 (+0.3) |
> > >
> > > In Table R6, SFT (V0) performs reasonably under direct generation (78.3 avg), but degrades sharply when self-correction is applied (44.2, −34.1). In contrast, our model remains stable (80.6 → 80.9, +0.3). Notably, SFT (V0) becomes even less stable than the base Qwen3-8B under self-correction, suggesting that fitting neutral outputs alone does not support reliable correction.
> > >
> > > **Table R7:** Bias injection recovery comparing SFT (V0) and SFT+SC (Ours).
> > >
> > > | Metric | SFT (V0) | SFT+SC | Δ |
> > > |---|---|---|---|
> > > | **BBQ (In-Dist., N=1000)** | | | |
> > > | Direct Gen. | 97.7% | 98.3% | +0.6 |
> > > | Injection + SC | 92.8% | 96.1% | +3.3 |
> > > | Aha! Rate | 29.9% | 33.6% | +3.7 |
> > > | **CrowS-Pairs (OOD, N=380)** | | | |
> > > | Direct Gen. | 59.4% | 67.4% | +8.0 |
> > > | Injection + SC | 56.1% | 61.6% | +5.5 |
> > > | Aha! Rate | 16.8% | 24.0% | +7.2 |
> > >
> > > Table R7 provides a more controlled view. On in-distribution BBQ, both models achieve similar direct generation performance (97.7% vs. 98.3%), indicating comparable learning of neutral output patterns. However, under bias injection, our model recovers more effectively (+3.3) and shows a higher Aha! rate (+3.7).
> > >
> > > On OOD CrowS-Pairs, the gap widens further: our model achieves higher direct accuracy (+8.0), stronger recovery (+5.5), and a substantially higher Aha! rate (+7.2). In contrast, SFT (V0) shows limited ability to detect and revise biased reasoning.
> > >
> > > **Takeaway.** Across both settings, the two models behave similarly in standard generation but diverge when correction is required. This suggests that the self-correction term contributes a distinct capability for detecting and revising biased reasoning, rather than simply improving the output distribution.
> > >
> > > We will include these results in the revision for clarity.
> > >
> > > We thank the reviewer again for the constructive feedback, which helped strengthen this analysis.

---

### Official Review · Reviewer_TYrE · 2026-03-13

**Soundness:** 3
**Presentation:** 3
**Significance:** 2
**Originality:** 3
**Overall Recommendation:** 4
**Confidence:** 4

**Summary:**

The background of this paper is that existing models may exhibit social bias (such as gender bias) during generation, which can lead to biased outputs for certain queries. To mitigate this issue, the paper proposes a SELF-DEBIAS strategy to debias social bias during the reasoning process. Unlike traditional intrinsic and external (self-reflection) debiasing methods, SELF-DEBIAS is a training-based strategy featuring three stages. The first stage uses a cold-start strategy with standard SFT to mitigate biased outputs during inference. In the second stage, SELF-DEBIAS employs a DPO-based strategy to allocate limited probability mass resources toward more unbiased outputs. In the third stage, online training data is generated from the policy model to reinforce unbiased behavior through learning from trajectories sampled from the policy itself. Experiments demonstrate that SELF-DEBIAS effectively mitigates injected bias across several datasets while maintaining model utility.

**Compliance With Llm Reviewing Policy:**

Affirmed.

**Final Justification:**

The authors have addressed my concern.

**Key Questions For Authors:**

1. What if you do not inject the biased context directly at the start of the reasoning process? It would be interesting to see the manifestation and impact of bias throughout the entire reasoning process.

2. Could you explain why model critique fails to mitigate bias and instead degrades performance? Does the critique prompt outline specific rubrics for bias detection?

3. Could you show the output differences between the trained model and the original base model when dealing with biased prompts?

4. Have you tried more advanced reasoning models to see whether they can perform self-debiasing or to assess their inherent resistance to social bias?

**Limitations:**

yes

**Strengths And Weaknesses:**

### Strengths

- The proposed SELF-DEBIAS is effective in mitigating social bias; it injects debiasing capabilities into the model through training while maintaining the model's utility.
- The proposed method is reasonable; the workflow, methodology, and analysis are clear. The entire paper is well-written and easy to read.
- The method is effective in mitigating social bias during generation.

### Weaknesses

- The paper focuses on mitigating social biases within the reasoning process; however, it lacks a formal definition of social bias, specifically regarding its category and scope. It is suggested to outline the specific types of social bias addressed in this paper and, based on this, analyze whether the method is generalizable across different bias categories.
- The comparison between SELF-DEBIAS and conventional self-reflection or model critique strategies is somewhat unfair, as SELF-DEBIAS requires updating the policy model while the other methods are prompt-based without training. Additionally, it is unexpected that both self-reflection and model critique degrade model performance. Have you tried using different or stronger models (other than the model itself) for the model critique?
- This paper investigates social bias within the reasoning process by injecting biased context at the start of the reasoning process, which is valid for preliminary studies. However, this process is somewhat artificial. When using the model in the wild, does it still exhibit explicit social bias without prompt injection? If not, it is important to reflect on the motivation of this setting: in what specific cases would the model receive such strong bias during generation?

---

> ### Author Rebuttal · Authors · 2026-03-30
>
> We sincerely thank you for your detailed comments and for acknowledging that our method is reasonable and effective.
>
> > **W1: Formal definition of social bias; generalizability across bias categories?**
>
> A: Our work addresses social biases across nine demographic dimensions (gender, race, age, religion, nationality, disability, socioeconomic status, sexual orientation, and physical appearance), as defined by our evaluation benchmarks. These manifest as (1) stereotype-driven reasoning errors in QA tasks (BBQ, UnQover) and (2) counterfactual inconsistencies in classification tasks (CEB). While SELF-DEBIAS is trained primarily on BBQ, it achieves consistent improvements across all Out-Of-Distribution (OOD) benchmarks (Table 1), confirming it learns generalizable debiasing capabilities rather than category-specific patterns. We will add this formal scoping to the revision.
>
> > **W2: Unfair comparison with prompt-based self-reflection/model critique (no training)?**
>
> A: We respectfully contend that the comparison is fair. As demonstrated in Table 2, all baseline methods were applied directly to our fully trained SELF-DEBIAS Iter2 model, ensuring the same optimized backbone and identical policy weights. The observed performance discrepancies therefore strictly isolate the impact of the intervention strategies alone.
>
> > **W3 & Q1: Artificial forced-prefix setting; does bias emerge naturally?**
>
> A: On 1,000 BBQ samples with standard generation (no injection), the base Qwen3-8B produced 89 errors, 29.2% of which involved stereotype-reliant reasoning:
>
> **Table R3: Natural bias analysis on 1,000 BBQ samples without injection.**
>
> | Metric                | Base  | Self-Debias (Ours) |
> | :-------------------- | :---: | :----------------: |
> | Accuracy              | 91.1% |       97.4%        |
> | # Errors              |  89   |    26 (70.8% ↓)    |
> | Chain-level Bias Rate | 29.2% |       23.1%        |
> | Step-level Bias Rate  | 9.3%  |        8.0%        |
>
> SELF-DEBIAS reduces natural errors by 70.8%, confirming that bias genuinely emerges without injection and our method effectively mitigates it.
>
> Regarding Q1, framework does not assume bias occurs at any fixed position. In Stage II, the truncation point t is randomly selected within the CoT sequence (Appendix B.1), so the model learns to detect and correct biases at arbitrary positions. The consistent improvements across diverse OOD benchmarks with varying bias patterns (Table 1) further validate this positional generalizability.
>
> > **Q2: Why does model critique fail and degrade performance?**
>
> A: Generic prompts fail to pinpoint specific biased steps within the reasoning chain, leading to indiscriminate overcorrection that compromises logical consistency, as illustrated in Figure 3 and Table 2.
>
> > **Q3: Output differences between trained and base model?**
>
> A: Please refer to Appendix C.4, which provides detailed case studies across BBQ, UnQover, CrowS-Pairs, CEB, ARC-C, and GSM8K. These illustrate how the base model rationalizes stereotypes or produces reasoning-answer mismatches, while SELF-DEBIAS triggers explicit self-verification steps to arrive at unbiased conclusions.
>
> > **Q4: Have you tried stronger external models for critique?**
>
> A: Stronger external models could indeed improve critique results, but introducing external dependencies inherently contradicts our core objective of achieving *intrinsic* self-correction independent of external oversight.

---

> > ### Author Rebuttal · Reviewer_TYrE · 2026-04-03
> >
> > Thanks for your rebuttal.

---

> > > ### Author Response · Authors · 2026-04-03
> > >
> > > Dear Reviewer TYrE,
> > >
> > > Thank you very much for your positive feedback and for your careful review of our rebuttal. We sincerely appreciate your acknowledgment that our responses have addressed all your concerns, as well as your recognition of our work.
> > >
> > > We are especially grateful for your updated evaluation following the rebuttal, and we truly appreciate your willingness to revise the score based on our clarifications and additional experiments. This means a lot to us.
> > >
> > > Thank you again for your valuable feedback and support.
> > >
> > > Best regards,
> > >
> > > Authors

---

### Official Review · Reviewer_Y4aS · 2026-03-13

**Soundness:** 3
**Presentation:** 3
**Significance:** 3
**Originality:** 3
**Overall Recommendation:** 5
**Confidence:** 4

**Summary:**

The authors address the problem of bias propogation in LLMs CoT- that if a social bias appears early in a reasoning chain, subsequent steps tend to rationalise it. Their technique involves both a fairness-aware DPO style optimisation of trajectories and online self-improvement via filtering. They show that this technique improves on fairness benchmarks while not regressing on reasoning capabilities.

**Compliance With Llm Reviewing Policy:**

Affirmed.

**Final Justification:**

This is an important and valuable piece of work with very useful results, and the authors have shown their approach to mitigating it generalises across LLM backbones & across synthetic and naturally occurring forms of bias propogation.

**Key Questions For Authors:**

- does the improvement (especially in inference time correction) generalise to other model families/still hold on a  stronger backbone, or with higher reasoning effort?
- how representative is the forced prefix bias-injection to bias propogation that emerges naturally - is there some way to quantify this?

**Limitations:**

yes

**Strengths And Weaknesses:**

Strengths:
- The problem being addressed is well motivated:  bias propagation is real and urgent & often overlooked in lieue of shorter-horizon measures of bias.
- A key finding/actionable result of this paper is that just inference time self-correction doesn't help, and actually worsens bias. This is an important result for the field at large, as its what ppl usually reach for as a fix
- The author's approach to addressing this: cold-start + fairness aware DPO at the suffix level is a principled technique that targets bias propogation rather than penalising entire trajectories -  and seems to improve the inference time self-correction on a Qwen backbone.
- Ablations are quite comprehensive and the authors do a good job of comparing with past work.

Weaknesses:
- Evaluation is limited to a single backbone (Qwen3-8b). I don't consider this to particularly weaken the stated claims, but it does limit their generalisabily. Showing results on one of the other baselines they use (llama 3 8b/deepseek distill 7b for e.g.) would strengthen the claims.

---

> ### Author Rebuttal · Authors · 2026-03-30
>
> We sincerely thank you for your positive assessment and insightful questions.
>
> > **W1 & Q1: Does the improvement generalize to other model families or stronger backbones?**
>
> A: Thanks for your insightful comment. To verify generalizability, we have conducted additional experiments applying our exact pipeline to Llama-3.1-8B-Instruct.
>
> **Table R1: Generalization results on Llama-3.1-8B-Instruct backbone.**
>
> | **Model Setting**          | **BBQ**  | **UnQ**  | **CEB-A** | **CEB-C** | **CEB-J** | **CrowS** | **ARC-C** | **GSM8K** | **Avg.** | **Δ** |
> | -------------------------- | -------- | -------- | --------- | --------- | --------- | --------- | --------- | --------- | -------- | ----- |
> | LLaMA-3.1-8B-Instruct      | 69.8     | 33.5     | 21.6      | 11.6      | 67.3      | 54.2      | 78.6      | 81.8      | 52.3     | -     |
> | + Self-Correction          | 50.2     | 57.8     | 6.9       | 8.0       | 29.1      | 51.0      | 71.9      | 67.2      | 42.8     | -9.5  |
> | Self-Debias SFT            | 96.8     | 99.5     | 66.5      | 64.2      | 70.5      | 68.2      | 92.9      | 86.2      | 80.6     | -     |
> | + Self-Correction          | 96.9     | 99.5     | 66.9      | 64.6      | 70.7      | 67.5      | 92.1      | 88.9      | 80.9     | +0.3  |
> | Self-Debias Offline        | 97.1     | 99.5     | 67.5      | 62.3      | 71.7      | 67.8      | **93.8**  | 86.7      | 80.8     | -     |
> | + Self-Correction          | 97.1     | 99.6     | 67.1      | 64.3      | 72.5      | 68.5      | 93.2      | 88.6      | 81.3     | +0.5  |
> | Self-Debias Online (Iter1) | 96.9     | 99.6     | 68.3      | 63.0      | 73.1      | 70.0      | 92.5      | 87.3      | 81.3     | -     |
> | + Self-Correction          | 97.0     | 99.5     | 67.2      | 63.9      | **73.5**  | 70.2      | 92.3      | **89.4**  | 81.4     | +0.1  |
> | Self-Debias Online (Iter2) | 97.0     | 99.5     | 67.1      | 65.8      | 72.1      | 71.2      | 93.1      | 87.6      | 81.4     | -     |
> | + Self-Correction          | **97.1** | **99.7** | **68.4**  | **65.8**  | 71.9      | **71.3**  | 91.1      | 87.0      | **81.5** | +0.1  |
>
> The results confirm that SELF-DEBIAS generalizes across model families: Llama-3.1 improves from 52.3 to 81.5 average score, with consistent gains at every stage. Notably, the base Llama model suffers severe degradation under self-correction (-9.5), while our trained model achieves stable positive improvements, mirroring the pattern observed on Qwen3-8B.
>
> > **Q2: How representative is the forced prefix bias-injection to bias propagation that emerges naturally?**
>
> A: Thank you for your valuable comment. The forced prefix is a diagnostic tool to isolate stereotype activation, which we confirm naturally occurs in practice. On 1,000 BBQ samples without any injection, the base Qwen3-8B model produced 89 errors, 29.2% of which involved stereotype-reliant reasoning chains:
>
> **Table R2: Natural bias analysis on 1,000 BBQ samples without injection.**
>
> | Metric                | Base  | Self-Debias (Ours) |
> | :-------------------- | :---: | :----------------: |
> | Accuracy              | 91.1% |       97.4%        |
> | # Errors              |  89   |    26 (70.8% ↓)    |
> | Chain-level Bias Rate | 29.2% |       23.1%        |
> | Step-level Bias Rate  | 9.3%  |        8.0%        |
>
> SELF-DEBIAS reduces natural errors by 70.8%, confirming that the capability learned via forced-prefix training generalizes effectively to naturally emerging biases. This is further validated by the strong direct generation results across all OOD benchmarks in Table 1.

---

> > ### Author Rebuttal · Reviewer_Y4aS · 2026-04-03
> >
> > Thanks to the authors for this response - the generalisation to the llama backbone & to naturalistic errors substantially alleviates any concerns I had remaining.

---

> > > ### Author Response · Authors · 2026-04-03
> > >
> > > Dear Reviewer Y4aS,
> > >
> > > Thank you very much for your positive feedback and for your careful review of our rebuttal. We sincerely appreciate your acknowledgment that our responses have addressed your concerns.
> > >
> > > We are particularly glad that the additional experiments and analyses have made the evaluation more comprehensive and strengthened the overall empirical support of our work.
> > >
> > > Thank you again for your valuable feedback and support.
> > >
> > > Best regards,
> > > Authors

---

### Decision · Program_Chairs · 2026-04-30

**Decision:**

Accept (regular)

**Comment:**

**Summary:** This paper addresses bias propagation in Chain-of-Thought reasoning, where an initial biased step leads subsequent reasoning to rationalize and amplify the bias. The authors propose Self-Debias, a three-stage framework: cold-start SFT to train both direct generation and self-correction capabilities, fairness-aware DPO with a Jain's Index regularizer operating at the trajectory suffix level, and online self-improvement via consistency filtering. Experiments on six fairness benchmarks and two reasoning benchmarks show substantial debiasing gains while preserving general reasoning ability, with results generalizing from Qwen3-8B to Llama-3.1-8B.

**Summary of reviews:** Scores are 2, 4, 4, and 5, with one reviewer raising from 3 to 4 after the rebuttal. Three reviewers praised the well-motivated problem formulation, the principled suffix-level DPO design, comprehensive ablations, and cross-model generalization. The central concern about whether the SFT gains reflect genuine self-correction or distribution fitting was raised by one reviewer, who then found the SFT(V0) ablation convincing and raised their score accordingly. The lowest-scoring reviewer did not engage with the rebuttal responses and instead introduced a new critique about the philosophical foundations of utility functions in preference learning, a concern that applies to the entire DPO/RLHF literature rather than to this specific submission.

**Assessment:** The paper tackles a meaningful gap: existing debiasing methods do not address how bias propagates through multi-step reasoning chains. The key finding that naive self-correction actually worsens bias is practically important. The three-stage framework is well-structured and the suffix-level DPO formulation is a sensible design choice over full-trajectory penalties. The rebuttal was strong. The Llama-3.1 cross-model transfer demonstrates that the approach is not tied to one backbone. The SFT(V0) ablation isolates the contribution of the self-correction stage from distribution fitting, and the bias injection recovery experiments provide an informative test of the self-correction mechanism under adversarial conditions. The concern about the small marginal gain at inference time is reasonable, but the bias injection experiment with its +3.3 recovery and higher self-initiated correction rate shows the mechanism does activate when needed. I have downweighted the lowest review, which shifted its evaluation criteria post-rebuttal without engaging with the authors' responses.